# Are Dairy Cow Replacement Decisions Economically Justified? Evidence from Swiss Farms

**DOI:** 10.3390/ani15162442

**Published:** 2025-08-20

**Authors:** Simon Schlebusch, Rennie Eppenstein, Daniel Hoop, Peter von Rohr

**Affiliations:** 1Agroscope, Tänikon 1, 8356 Ettenhausen, Switzerland; 2Braunvieh Schweiz, Chamerstrasse 56, 6300 Zug, Switzerland; 3FiBL, Ackerstrasse 113, 5070 Frick, Switzerland; rennie.eppenstein@fibl.org; 4Qualitas AG, Chamerstrasse 56, 6300 Zug, Switzerland; peter.vonrohr@qualitasag.ch

**Keywords:** dairy cattle, cow replacement, culling decisions, longevity, economic modeling

## Abstract

Dairy farmers regularly decide whether to keep or replace a cow in their herd. This replacement decision has a big impact on animal welfare, the environment, and farm income. Many people believe that cows are often removed too early, before they have reached their full productive potential. In this study, we looked at detailed records from 29 Swiss dairy farms over five years to understand how these decisions are made in real life. We used a tool that calculates the economic value of each cow based on her health, fertility, and milk production. We then compared the farmers’ decisions to what the tool suggested. We found that most farmers made good economic choices, but that some cows—especially younger ones—were removed earlier than necessary. While these early removals sometimes make sense financially, they are not ideal for the environment or for the well-being of the animals. On average, these suboptimal decisions resulted in an economic loss of 161 ± 164 CHF per farm per month and 1.55 ± 1.58 CHF per cow per month; also, economic loss from retaining unprofitable cows was about three times greater than from premature culling. This study helps explain how farmers make these decisions and shows how future improvements in health and breeding could reduce early culling and support more sustainable farming.

## 1. Introduction

The decision of whether to retain or replace a dairy cow is one of the most frequent and economically significant challenges in herd management. Typically, 20% to 40% of cows are replaced annually [1,2,3,4,5]. In Switzerland, where the dairy cow population exceeds 500,000 head, this corresponds to over 100,000 replacements per year [6]. Each replacement event involves not only the loss of a productive animal but also substantial investment in raising or purchasing a heifer [7,8,9,10].

Despite this economic importance, the average productive lifespan of dairy cows remains short—typically 2.5 to 4 parities worldwide, including in Switzerland [1,11,12]. This is one to two parities shorter than the suggested biological and economic optimum of five to six parities [2,9,13,14], implying a potential loss of efficiency due to premature culling.

Many theoretical models suggest that extending cow longevity would improve profitability and sustainability. However, the practical relevance of these models is uncertain. Few empirical studies investigate how farmers actually make replacement decisions or whether they align with calculated economic optima. Exceptions include [7,13], who used farm accounting data. But, overall, there is a notable gap between theoretical recommendations and on-farm behavior.

The Markov chain approach is widely recognized for its ability to model dynamic biological and economic processes in livestock systems, particularly replacement decisions, by incorporating transitions between health, reproductive, and production states over time [15,16,17]. Its integration here not only ensures robust economic evaluation but also allows explicit linking replacement decisions to sustainability metrics, as it quantifies the potential to extend productive lifespans and reduce environmental impacts per unit of milk.

The present study addresses this gap by combining detailed replacement data from 29 Swiss dairy farms with a bio-economic cow value model [18]. Over a 5-year period, we examine each culling decision and compare it to the theoretical optimum. The aim is to assess whether farmers make economically rational decisions, and to what extent early replacement is truly a mistake—or, instead, a rational response to biological or economic constraints.

## 2. Materials and Methods

### 2.1. Study Design and Data Sources

This study evaluated the economic justification of dairy cow replacement decisions by analyzing herd-level data from 29 Swiss farms over a five-year period (2018–2023). The farms were selected to represent a diverse range of characteristics, including breed composition, production region, herd size, milk yield, production system (organic or conventional), and barn type. These farms were originally recruited as part of the Swiss research project Nutzungsdauer Schweizer Milchkühe, which focuses on improving dairy cow longevity.

Each participating farm provided full access to its herdbook records. These included 305-day lactation data such as yield, fat, and protein content; insemination and pregnancy records; detailed culling dates and associated reasons; and monthly milk recording data. These biological and management records were combined with farm-specific economic parameters—including milk prices, feed costs, and veterinary expenses—to compute monthly profitability measures and to simulate cow-level replacement scenarios under a standardized economic framework.

### 2.2. Cow Value Model Overview

To evaluate whether a cow should be retained or replaced, we used a bio-economic simulation model that estimates the net value of each cow within its herd context. The model calculates the “cow value” as the difference between the expected monthly profitability of the cow and that of a replacement heifer.$$CowValue={\bar{MCM}}_{cow}-{\bar{MCM}}_{replacement}$$

Here, $\bar{MCM}$ refers to the average monthly contribution margin, which reflects the cow’s profitability over her remaining productive lifespan. A positive cow value suggests the cow is more profitable than a replacement and should be kept. A negative cow value implies replacement is economically preferable.

### 2.3. Markov Chain Modeling of Cow States

The model uses a discrete-time Markov chain to simulate the transitions between biological and economic states in a cow’s lifecycle. Each state is defined by a combination of the following:Lactation number (parity);Month in milk;Month in pregnancy.

Using historical data, the model estimates transition probabilities for moving between physiological and reproductive states from one month to the next. Two large datasets were used to compute these probabilities. The milk recording data, collected from 2010 to 2018, included 1,016,428 cows with detailed records on calvings, milk test days, and lactation numbers, and was used to model transitions in lactation stage and milk month. A second dataset, covering insemination records for 1,282,749 cows over the same period, was used to estimate the monthly probability of changes in pregnancy status.

Starting from a cow’s observed state—defined by her current parity, month in milk, and pregnancy status—the model uses iterative Markov simulations to estimate the probabilities of transitioning through all future states, the expected productive lifespan until culling, and the cow’s total lifetime economic value. The Markov chain continues until the cow reaches a terminal state (i.e., culling). Based on these projections, the model calculates the expected monthly contribution margin (MCM) over the remaining life of the animal. This modeling approach builds on the framework introduced by [18], which applied a similar economic evaluation to the context of Swiss dairy farms.

### 2.4. Monthly Contribution Margin (MCM) Calculation

The monthly contribution margin (MCM) represents the net profitability of a cow in a given month. It is calculated as the difference between revenues and costs:$${MCM\;}_{t}=\;{Revenue}_{t}\;-{Cost}_{t}$$

Revenue consists of three main components: milk income, calf income, and culling income. Milk income is based on the cow’s milk yield, adjusted for fat and protein content, with yield estimated using herd-specific lactation curves fitted with the Wood function [19]. Calf income reflects the revenue from selling calves shortly after birth, while culling income accounts for the carcass value received when cows are removed from the herd.

Costs are composed of feed, veterinary, insemination, and replacement expenses. Feed costs depend on milk yield, milk composition, live weight, and pregnancy status. Veterinary costs are modeled as a fixed monthly average per cow, while insemination costs are incurred for cows that are not pregnant by the third month postpartum. Replacement costs apply when a cow is culled and a new heifer is introduced into the herd.

Economic parameters used in the model were based on farm data and reflect Swiss production conditions. Across the 29 farms, the average milk price was 0.797 ± 0.069 CHF/kg, veterinary costs averaged 977.20 ± 1080.85 CHF per cow per year, and insemination costs were 66.80 ± 20.85 CHF per service. The average cost of replacement heifers was 3123.33 ± 404.50 CHF. Feed costs were based on the prices of roughage (0.409 ± 0.12 CHF/kg DM) and concentrate feed (0.88 ± 0.26 CHF/kg DM). The average slaughter price was 3.69 ± 0.73 CHF/kg carcass weight, and calf income was calculated using an average price of 7.33 ± 2.36 CHF/kg live weight. Where farm-specific values were missing, the corresponding dataset average was used to maintain model consistency. All monetary values are expressed in Swiss francs (CHF), reflecting the local currency used by participating farms. For reference, the average 2023 exchange rates were 1 CHF ≈ 1.02 EUR ≈ 1.10 USD.

The average monthly contribution margin for each cow is calculated over her expected remaining productive lifetime, as predicted by the Markov chain model. For each individual cow, the model projects monthly MCM values based on her predicted life path. The average MCM is then determined by summing these monthly values and dividing by the total number of productive months.$$\bar{MCM}=\frac{1}{n}\;\;\sum_{t=1}^{n}{MCM}_{t}$$

This is computed for both the actual cow, based on her current state, and a hypothetical replacement heifer, expected to perform at the herd average starting in first parity.

### 2.5. Decision Rule and Integration of Farm Data

To assess the economic justification of each culling decision, the model calculates the cow value at the last observed point in time for every cow. From this point forward, the calculated life expectancy is defined as the predicted number of additional productive parities, as estimated by the Markov model based on the cow’s current state. This metric does not directly indicate whether a cow will be culled, but rather reflects the model’s projection of her remaining productive lifespan. The decision rule is straightforward: if the cow value is greater than zero, the cow is considered more profitable than her replacement and should therefore be retained. Conversely, if the cow value is less than zero, the replacement heifer is expected to be more profitable, and the cow should be culled. This rule was applied monthly to all cows across the 29 farms using the most recent herdbook data. Since year-round calving is common in the Swiss dairy farming sector, seasonal effects are less pronounced than in more seasonally structured systems. Therefore, no additional seasonal adjustments were applied beyond those inherently captured by the model. For each animal, the evaluation combined information on her current state (including parity, days in milk, and pregnancy status), herd-specific economic parameters, her Markov-predicted life trajectory, and the observed outcome—whether the cow was ultimately culled or retained. By comparing the model’s recommendation to the farmer’s actual decision, each cow was classified into one of four scenarios, allowing for an assessment of decision quality across the dataset as presented in Table 1.

## 3. Results

### 3.1. Descriptive Statistics of the Study Farms

The study included 29 Swiss dairy farms, representing a range of production conditions and breeds; across the entire 5-year period, the dataset included 3003 individual cows. The predominant breeds were Brown Swiss (BV), Holstein (HO), Original Brown (OB), Swiss Fleckvieh (SF), and Montbéliard (MO). Across the entire 5-year period, the number of individual cows recorded per farm ranged from 37 to 237, with a mean of approximately 104 cows. These figures reflect the total number of different cows observed per farm, not the average herd size at any single point in time.

The replacement rate varied substantially, ranging from 3.6% to 38.6%, with an average of approximately 24.5 ± 6.9%. This variability reflects differences in herd size, production system, culling strategy, and breeding objectives.

Table 2 summarizes the dominant breed, total number of cows observed per farm, and replacement rates over the study period. In addition to these descriptive metrics, Table 2 also includes key economic indicators from the cow value model analysis, discussed below.

### 3.2. Economic Evaluation of Replacement Decisions

To evaluate the economic rationality of replacement decisions, each cow’s final state was assessed using the cow value model. A positive cow value indicates that the cow was more profitable than her replacement and should have been retained; a negative value suggests that the cow should have been replaced.

These categories allowed us to quantify economically justified and unjustified decisions at the herd level.

Across all 29 farms,

Cows that were culled despite having a positive cow value accounted for a cumulative economic loss of 1101 CHF.Cows that were retained despite a negative cow value resulted in a larger loss of 3557 CHF.

These losses were unevenly distributed across farms. Some showed mostly optimal decisions, while others had higher rates of suboptimal replacements. When averaged across all farms, the total economic loss from replacement decisions amounted to 4608 CHF, or 161 ± 164 CHF per farm per month. On a per-animal basis, this equates to 1.55 ± 1.58 CHF per cow per month—a seemingly small number that adds up significantly over time.

These results are summarized in the lower half of Table 2 under the columns “Replaced” and “Not Replaced,” which show the cumulative cow values lost due to premature culling and delayed replacement, respectively.

### 3.3. Patterns and Reasons for Cow Replacement

To better understand the patterns underlying suboptimal replacement decisions, we examined cow value, age, culling status, and stated reasons for removal. Figure 1 illustrates the relationship between cow value and age in parities for one example herd. Each point represents an individual cow, with color indicating life expectancy (in parities) and shape indicating culling status (circles for retained cows, triangles for culled ones). Most cows with positive cow values were retained, as expected. However, several cows with positive value were culled prematurely, indicating potential economic loss. Notably, no cows with negative values remained in the herd, suggesting that farmers are generally more cautious about retaining underperforming cows than about culling potentially profitable ones.

To complement this analysis, Table 3 summarizes the reasons for cow replacement across parities for all 553 culled animals. Replacements occurred most frequently in first and second parity, together accounting for 36% of all removals. This is notable, as cows culled early in life often have not yet recovered the investment made in raising them. The three leading causes of replacement were fertility issues (26.4%), udder health problems (22.6%), and inadequate performance (9.8%). A non-negligible share (13.7%) of cows were sold alive, particularly in early parities, which may reflect changes in herd strategy or voluntary removal.

These patterns confirm that both biological and management factors drive replacement decisions, and that economic misalignments—such as culling cows with positive value—are not uncommon, particularly in younger animals.

### 3.4. Economic Impact of Health and Fertility-Related Culling

To assess the financial implications of specific health problems, we analyzed the cumulative cow value associated with cows culled due to fertility issues and udder health problems—the two most frequent reasons for replacement in this study.

Across all 29 farms,

Cows culled for fertility problems (n = 146) accounted for a cumulative loss of 1558 CHF, as 28 of these cows had a positive cow value at the time of culling.Cows removed due to udder health issues (n = 125) led to a smaller but still notable loss of 858 CHF, with 19 cows culled despite being more profitable than their replacement.

These numbers highlight that not all biologically justified culling is economically optimal. Even in the presence of health problems, some cows had sufficient remaining profitability to justify continued retention—particularly when replacement costs were high or future performance was expected to recover.

The findings suggest that systematic use of an economic decision tool could help farmers distinguish between cows whose removal is both biologically and economically justified and those for whom retention would yield higher long-term returns.

### 3.5. Scenario-Based Assessment of Key Culling Reason

In addition to the empirical analysis of cow-level outcomes, we conducted a scenario-based economic assessment using the cow value model to better understand how specific biological states influence profitability. This simulation focused on three major culling reasons: fertility problems, inadequate performance, and udder health issues (as introduced in Section 2.4). The goal was to quantify how these factors affect cow value under otherwise identical herd and market conditions.

Table 4 presents the results of this analysis, showing the economic consequences of delayed pregnancy, reduced milk yield, and a one-time health event (e.g., mastitis or claw disorder). For each combination of month in milk and pregnancy status, we compared three scenarios:The baseline cow (herd average performance);A cow with 10% lower milk yield;A cow expected to incur a costly health event (800 CHF) in the following month.

The results highlight the strong economic impact of reproductive performance. For example, a cow in her fifth month in milk gains 64 CHF in cow value by becoming pregnant, compared to remaining open. Early pregnancy consistently adds value, while each additional open month erodes profitability. Fertility also partially offsets the negative effects of lower milk yield—but not entirely.

In contrast, cows facing a single expensive health event often struggle to recover their financial viability. Depending on their stage of lactation and pregnancy status, these losses range from −97 to −231 CHF, and are especially severe for low-producing animals. In such cases, replacement may be the more profitable decision, even if the cow is otherwise fertile.

## 4. Discussion

This study used a dynamic cow value model to evaluate the economic rationality of replacement decisions on 29 Swiss dairy farms. On average, farmers made economically sound choices regarding when to cull or retain individual cows, though there was considerable variation across farms. The economic loss due to suboptimal culling ranged from 10 to 745 CHF per farm per month, with a herd-level average loss of 1.55 ± 1.58 CHF per cow per month. Notably, the economic loss from retaining cows too long (1.18 CHF per cow and month) was approximately three times greater than that from premature culling (0.33 CHF per cow and month). These findings suggest that farmers tend to be more cautious about keeping underperforming cows than they are about culling cows too early, and that most replacement decisions are economically justifiable.

These results appear to contrast with earlier research that emphasized low longevity in dairy herds as a persistent problem. Previous studies suggest that the optimal lifetime of a dairy cow is typically five to six parities [2,9,13,14]. However, the current findings indicate that short productive lifespans are not necessarily a result of irrational economic behavior. Rather, they may reflect unavoidable biological constraints or risk management strategies at the farm level. This interpretation aligns with recent work by [2,9,12,20], who argue that culling decisions are often driven by fertility problems, udder health issues, or other clinical events. Only [21] reported that low performance, rather than health or fertility, was the primary culling factor—differing from our findings and the broader literature.

A more nuanced question raised by our results is why so many cows are culled during their first or second parity, even though they have not yet recovered the investment made in their rearing. Contrary to the assumption that early culling reflects poor decision-making, our findings suggest that many early removals are economically justified — a novel insight that reframes how longevity targets should be interpreted. While some of these early culls are economically justified—possibly due to high carcass prices offsetting replacement costs—they still reflect a loss of potential lifetime productivity. Importantly, such decisions may be economically rational in the short term but ecologically and ethically suboptimal from a broader sustainability perspective [1,8]. Replacing a heifer after only one or two lactations increases the carbon footprint per liter of milk and reduces overall animal welfare due to shortened lifespans.

Our scenario-based analysis (Table 4) provides additional insight into the underlying drivers of cow value. Fertility was the most influential factor: cows that conceived earlier were consistently more valuable than those that did not. Health events, such as mastitis or claw disorders, also had a large negative economic effect, especially when occurring in cows with below-average milk yield. These findings suggest that the optimal replacement decision is not only a matter of the current cow state, but also of how and why cows deteriorate over time. A cow in poor condition may justifiably be culled today—but avoiding that decline in the first place would be the more desirable outcome.

Taken together, the findings reinforce that economically optimal culling is not synonymous with optimal biological or ethical outcomes. A more sustainable approach would aim to reduce the frequency of economically justified removals by improving fertility management, health monitoring, and preventive care earlier in the cow’s productive life. These results have direct implications for agricultural policy and farmer support schemes. Encouraging the adoption of decision-support tools, either through extension, could improve alignment between economic, ecological, and welfare outcomes. Additionally, breeding programs and herd health plans could be tailored to reduce the incidence of early culling, thereby contributing to national sustainability goals and improving economic resilience at the farm level.

There are several limitations to our study. First, the cow value estimates depend on assumptions regarding future milk yield, reproductive success, culling probabilities, and market conditions. These may not fully reflect actual outcomes on individual farms. Second, the reasons for culling were extracted from farm records and may be incomplete or inconsistently defined. Third, our model assumes that replacement heifers perform at the herd average, which may not hold in herds with genetic variation or selective replacement strategies. Finally, the relatively small sample size of 29 farms limits the generalizability of our findings across broader regions or production systems. Future research could build on these findings by developing and testing integrated management frameworks that reduce the frequency of economically justified removals through targeted improvements in fertility management, proactive health monitoring, and preventive care early in the cow’s productive life. Such work could also address the current study’s limitations by incorporating farm-level genetic data, validating model predictions with prospective longitudinal monitoring, and expanding the dataset to include a wider range of production systems and geographical regions.

## 5. Conclusions

The results show that, on average, farmers make economically sound cow replacement decisions. Nonetheless, frequent early culling—particularly in first-parity cows—remains suboptimal from an ecological, animal welfare, and sustainability perspective. Improving animal health and fertility is therefore essential for reducing premature culling and extending productive lifespans. However, if fertility, health, or performance limitations are genetically determined and manifest in the first parity, farmers have limited short-term options. In such cases, long-term progress depends on adjustments to breeding strategies aimed at improving resilience and lifetime performance. We recommend the systematic integration of economic decision-support tools into dairy herd management, as these can help farmers identify and avoid economically suboptimal culling decisions, thereby supporting profitability, animal welfare, and environmental sustainability

## Figures and Tables

**Figure 1 animals-15-02442-f001:**
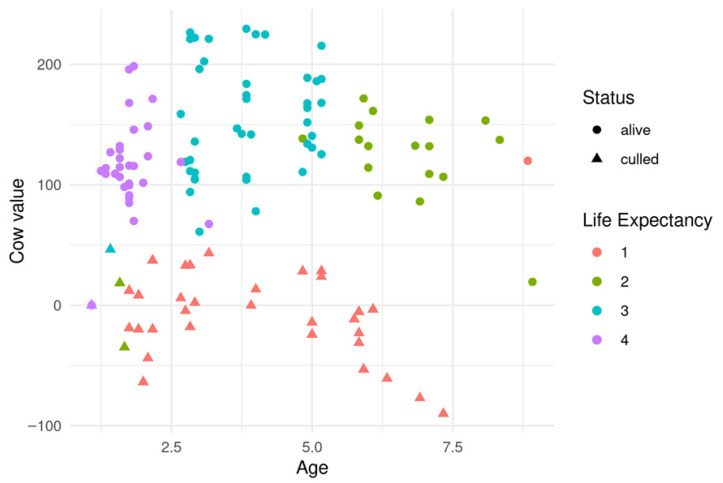
Cow value vs. age of the cow for one example herd. The *x*-axis shows the age in parities, and the *y*-axis shows the cow value in CHF. Colors represent the cow’s life expectancy (in parities), while shapes represent status (circle = alive, triangle = culled). Cows above the 0 CHF line are more valuable than their replacement; those below it are less valuable. While some cows with positive value were culled, no cows with negative value remained in the herd.

**Table 1 animals-15-02442-t001:** Classification of cow replacement decisions based on cow value and culling status. Each cow is assigned to one of four categories depending on whether she was culled or retained, and whether her cow value was positive (indicating she should have been retained) or negative (indicating she should have been culled). Suboptimal decisions occur when cows are either replaced too early or retained too long.

	Cow Culled	Cow Retained
Cow Value > 0	Suboptimal: Replaced Too Soon	Optimal: Retained
Cow Value < 0	Optimal: Correctly Culled	Suboptimal: Retained Too Long

**Table 2 animals-15-02442-t002:** Summary of farm-level data, including dominant breed, total number of individual cows recorded during the 5-year observation period, and replacement rate (%). The table also includes economic results from the cow value model: the “Replaced” column reports the cumulative cow value (in CHF) of cows that were culled despite being economically profitable (i.e., replaced too early), while the “Not Replaced” column shows the cumulative cow value (in CHF) of cows that were retained despite being economically unprofitable (i.e., retained too long).

ID	Breed	Replaced Cow Is Culled (in CHF)	Not Replaced Cow Is Alive (in CHF)	Number of Cows	Replacement Rate (%)
1	BV	−380.77	0	138	25.4
2	BV	0	−217	72	31.9
3	OB	0	−746	72	25.0
4	BV	10	0	40	20.0
5	BV	0	−11	74	17.6
6	BV	0	−100	89	28.1
7	BV	0	−209	45	33.3
8	BV	0	−13	55	3.6
9	OB	0	−84	37	24.3
10	BV	−25	−28	48	22.9
11	BV	−127	0	108	27.8
12	BV	0	−165	120	20.8
13	HO	−75	0	237	27.4
14	HO	0	−220	134	25.4
15	BV	0	−29	44	20.5
16	HO	−7	−178	167	22.8
17	HO	−28	−273	164	26.2
18	MO	0	−31	77	11.7
19	MO	0	−258	139	20.1
20	HO	0	0	68	26.5
21	HO	−18	−44	139	17.3
22	HO	−115	−40	129	34.1
23	MO	0	−502	126	22.2
24	HO	−121	0	236	38.6
25	SF	−169	0	63	20.6
26	SF	−24	0	94	26.6
27	HO	0	−30	120	23.3
28	BV	0	−221	52	32.7
29	BV	0	−159	116	24.1
Sum		−1101	−3557	3003	

**Table 3 animals-15-02442-t003:** Reasons for cow replacement by parity across all 553 culling events recorded during the 5-year observation period. Rows represent specific reasons for removal, while columns indicate the parity at which the cow was culled. The final columns show the total number and share (%) of replacements for each reason. Fertility problems, udder health issues, and inadequate performance were the most common causes of replacement. Early parities (1 and 2) accounted for the highest proportion of removals.

Parity	1	2	3	4	5	6	7	8	9	10	11	12	Total	Share
Calving issues	2	2	2	5	2	1	1	0	0	0	1	1	17	3.1%
Others	11	8	5	3	2	1	4	0	2	2	0	1	39	7.1%
Respiratory disease	1	2	0	0	0	0	0	0	0	0	0	0	3	0.5%
Udder health	15	9	15	21	17	24	8	7	1	3	3	2	125	22.6%
Fertility	22	17	21	22	17	22	12	11	1	0	0	1	146	26.4%
Claw issues	6	3	6	8	3	5	3	2	3	1	1	0	41	7.4%
Slow milking	1	0	2	0	0	0	0	0	0	0	0	0	3	0.5%
Sold alive	39	13	13	3	4	2	2	0	0	0	0	0	76	13.7%
Metabolic disease	2	2	2	2	1	1	0	1	0	0	0	0	11	2.0%
Accidents/injuries	5	8	6	5	3	4	1	0	1	0	1	0	34	6.1%
Inadequate performance	19	11	9	5	4	3	1	2	0	0	0	0	54	9.8%
Somatic cell count	0	2	2	0	0	0	0	0	0	0	0	0	4	0.7%
Total replaced	123	77	83	74	53	63	32	23	8	6	6	5	553	
Share of total replaced cows (%)	22	14	15	13	10	11	6	4	1	1	1	1	100	

Note: Sold alive refers to cows removed for reasons unrelated to health (e.g., genetic value, herd restructuring). While these sales generate revenue, they may still represent economically suboptimal decisions if the cow’s projected future value exceeded the net sale benefit.

**Table 4 animals-15-02442-t004:** Scenario-based evaluation of cow value under different biological and economic conditions. Each row represents a cow state defined by month in milk and pregnancy status. The “Cow value” column shows the estimated economic value (CHF) for a cow with average herd performance. The “Cow value with lower milk yield” shows the value assuming 10% below-average milk production. The “Cow value with mastitis” reflects the value of a cow expected to incur a one-time health cost of 800 CHF (e.g., mastitis or claw disorder) in the following month. Values are calculated using the default model assumptions and reflect average conditions across the 29 herds.

Parity	Month in Milk	Month in Pregnancy	Cow Value in CHF	Cow Value with Lower Milk Yield in CHF	Cow Value with Mastitis in CHF
1	2	0	−4	−48	−128
1	3	0	−12	−56	−135
1	4	0	−26	−69	−146
1	5	0	−41	−84	−159
1	6	0	−57	−100	−173
1	7	0	−76	−118	−189
1	8	0	−93	−135	−204
1	9	0	−110	−151	−219
1	10	0	−123	−164	−231
1	3	1	32	−14	−97
1	4	1	28	−17	−100
1	5	1	23	−22	−104
1	6	1	19	−27	−108
1	7	1	15	−31	−111
1	8	1	11	−34	−114
1	9	1	9	−36	−116

## Data Availability

The authors do not have permission to share data or the source code of the model.

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
