# Peer review of "Are Dairy Cow Replacement Decisions Economically Justified? Evidence from Swiss Farms"

_animals, 2025, doi:10.3390/ani15162442_

Round 1
Reviewer 1 Report
Comments and Suggestions for Authors
Notes on: Are dairy cow replacement decisions economically justified? 2 Evidence from Swiss farms
Thank you for the opportunity to read your work. My comments will be in “real time”; that is, I am writing them as I am reading and some questions that I have may be answered further in the text.
*** Normally, this is where I put line-by-line grammatical corrections. I have none for this paper. I honestly cannot remember the last article I reviewed in which there weren’t typos or improper grammar.
I found this read interesting and an important contribution to the literature. Although from an animal ethics perspective, reducing cows to "makes money" versus "costs money" is a bit troublesome for me, farming is a business after all. The clear cut decision making rubric is appropriate as a business strategy.
In fact, I would encourage the authors to develop a concept introduced in this paper for future examination. Notably, “A more sustainable approach would aim to reduce the frequency of economically justified removals by improving fertility management, health monitoring, and preventive care earlier in the cow's productive life.” I would encourage the authors, in a future paper, to examine what this sustainable approach would look like.
Author Response
Comment: I would encourage the authors to develop a concept introduced in this paper for future examination. Notably, “A more sustainable approach would aim to reduce the frequency of economically justified removals by improving fertility management, health monitoring, and preventive care earlier in the cow's productive life.” I would encourage the authors, in a future paper, to examine what this sustainable approach would look like.
Response:
We thank the reviewer for this encouraging feedback and for highlighting the importance of exploring sustainable approaches in greater detail. While this topic is beyond the scope of the current analysis, we agree that it represents a valuable direction for future research. Accordingly, we have added a sentence to the Discussion to emphasise that future work could investigate practical strategies for implementing such sustainable approaches at the farm level.
Change in manuscript:
Added to the Discussion (lines 375-379):
“Future research could build on these findings by developing and testing integrated management frameworks that reduce the frequency of economically justified removals through targeted improvements in fertility management, proactive health monitoring, and preventive care early in the cow’s productive life.”
Reviewer 2 Report
Comments and Suggestions for Authors
The paper describes a very interesting aspect of dairy herd management and the fact that it is based on real data increases its value.
There are few points that need to be clarified:
Line 156: how many cows were considered?
Table 2 and in all the paper: I suggest reporting values in $ or € to facilitate the readers
Table 2 “replaced cows” based on the assumption and on what reported, these values must be negative (otherwise the sum of 4,608 CHF /line 193 is not justified).
Lines 179-182 The concept is repeated many times in the paper, please avoid too many repetitions.
Figure 1: the cows’ life expectancy is unclear to me. I suggest adding the definition to the M&M section. If means that these cows should be culled, it could be simplified in “cows to be culled”.
Table 3: among the reasons for replacement there is also “sold alive”. This sounds like these cows were sold for their performances or for their genetic value. If this is correct, in my opinion these cows cannot be included in this table and in the economic evaluation, since these are revenues and their “exit” don’t depend on the presence of disease or other reasons. Please modify and/or comment.
Author Response
Comment 1: Line 156: how many cows were considered?
Response:
We thank the reviewer for pointing out this omission. We have now specified the exact number of cows included in the analysis.
Change in manuscript:
Results (189-190) reads:
«Across the entire 5-year period, the dataset included 3,003 individual cows»
Comment 2: Table 2 and in all the paper: I suggest reporting values in $ or € to facilitate the readers.
Response:
We appreciate the reviewer’s suggestion. As the study was conducted entirely in Switzerland, all economic data were originally recorded and analyzed in Swiss francs (CHF), which is also the standard currency used by the participating farms. Using CHF ensures consistency with the actual prices, costs, and values faced by farmers during the study period, and avoids potential distortions due to currency fluctuations. For the benefit of international readers, we have now added a brief statement in the Methods noting the approximate average 2023 exchange rates to EUR and USD.
Change in manuscript:
Added to Materials and Methods (149-151):
“All monetary values are expressed in Swiss francs (CHF), reflecting the local currency used by participating farms. For reference, the average 2023 exchange rates were 1 CHF ≈ 1.02 EUR ≈ 1.12 USD.”
Comment 3: Table 2 “replaced cows” based on the assumption and on what reported, these values must be negative (otherwise the sum of 4,608 CHF /line 193 is not justified).
Response:
We thank the reviewer for pointing out this inconsistency. We agree that the “Replaced” column in Table 2 should display negative values to represent economic losses, consistent with the “Not replaced” column. We have corrected these values.
Change in manuscript:
Updated Table 2: all “Replaced” values now shown as negative.
Comment 4: Lines 179–182: The concept is repeated many times in the paper, please avoid too many repetitions.
Response:
We agree and have removed redundant wording in the Results section (215-216). The text is now more concise without loss of meaning.
Change in manuscript:
Original:
Cows were categorized into four possible scenarios based on whether they were culled or retained and whether their cow value was positive or negative. These categories allowed us to quantify economically justified and unjustified decisions at the herd level.
Revised:
These categories allowed us to quantify economically justified and unjustified decisions at the herd level.
Comment 5: Figure 1: the cows’ life expectancy is unclear to me. I suggest adding the definition to the M&M section. If means that these cows should be culled, it could be simplified in ‘cows to be culled’.
Response:
We thank the reviewer for this suggestion. We have now added a definition of “life expectancy” in the
Change in manuscript:
Materials and Methods section (162-166):
To assess the economic justification of each culling decision, the model calculates the cow value at the last observed point in time for every cow. From this point forward, the calculated life expectancy is defined as the predicted number of additional productive parities, as estimated by the Markov model based on the cow’s current state. This metric does not directly indicate whether a cow will be culled, but rather reflects the model’s projection of her remaining productive lifespan.
Comment 6:
Table 3: among the reasons for replacement there is also “sold alive”. This sounds like these cows were sold for their performances or for their genetic value. If this is correct, in my opinion these cows cannot be included in this table and in the economic evaluation, since these are revenues and their “exit” don’t depend on the presence of disease or other reasons. Please modify and/or comment.
Response:
We appreciate the reviewer’s observation. “Sold alive” includes cows removed from the herd for reasons unrelated to health problems, such as herd size adjustments, genetic improvement, or changes in breeding strategy. While these sales generate revenue, they still represent an exit from the herd and may, in some cases, be economically suboptimal if the cows had a positive cow value at the time of sale. From an economic optimisation perspective, such sales can still lead to foregone future profitability and therefore merit inclusion in the economic evaluation. For transparency, we have clarified in the footnote to Table 3 that “sold alive” cows were evaluated in the same way as other culling categories, but that the underlying reasons are not health-related.
Change in manuscript:
Added to Table 3 footnote, Results section (266-268):
Note: Sold alive refers to cows removed for reasons unrelated to health (e.g., genetic value, herd restructuring). While these sales generate revenue, they may still represent economically suboptimal decisions if the cow’s projected future value exceeded the net sale benefit.
Reviewer 3 Report
Comments and Suggestions for Authors
Dear authors,
I congratulate you for choosing this research topic. It is important both for the progress of scientific research in farm management and for its practical application in improving sustainability and animal welfare. The effort you have put into collecting and integrating data from a diverse sample of Swiss farms, as well as the use of an advanced bio-economic model, are significant contributions.
However, I have a few observations and recommendations to improve the manuscript:
The Abstract and Simple Summary are clear and concise, but they could include key numerical values from the results (average losses per farm, % of suboptimal decisions) to make it easier for the reader to connect with the data.
The Introduction is well argued and highlights the knowledge gap addressed by the study. The paper proposes an empirical approach, which adds value by combining modelling with practical observations.
I suggest reinforcing the scientific justification for using the Markov model, adding more references about the validity of this method in replacement decisions. It would also be useful to explain earlier in the text its relevance for sustainability (alongside the economic relevance).
Materials and Methods include the data sources, variables used, the structure of the model, the way cow value and MCM are calculated, as well as the decision classification.
However, the economic parameters used could be described in more detail (actual values for veterinary costs, average milk prices, etc.).
It might also be necessary to clarify any seasonal adjustments made in the model.
The Results are logically structured, and the data are presented objectively, without exaggerations. Economic losses are quantified and decision patterns are identified.
I think some aggregated values could be complemented with measures of dispersion (SD, IQR).
The Discussion connects the results to the existing literature and critically addresses the difference between the economic optimum and the ecological/ethical one.
The implications for agricultural policies or farmer support schemes could be discussed in more detail.
The mention of limitations is adequate. The paper would be improved by discussing how these could be overcome in future research.
The Conclusions are well formulated and consistent with the presented data, but you could explicitly add a recommendation to systematically integrate economic decision-support tools in farms.
I congratulate you for the quality of the research and for the contribution you bring to understanding the economic and biological mechanisms that drive dairy cow replacement decisions. I believe your study can be a reference for future research and for optimising practices in this field.
Author Response
Comment 1: The Abstract and Simple Summary are clear and concise, but they could include key numerical values from the results (average losses per farm, % of suboptimal decisions) to make it easier for the reader to connect with the data.
Response:
We thank the reviewer for this suggestion. We have now added key numerical values to both the Abstract and the Simple Summary to provide the reader with a clearer overview of the main findings.
Change in manuscript:
Simple Summary(21-24)
On average, these suboptimal decisions resulted in an economic loss of 161 ± 164 CHF per farm per month and 1.55 ± 1.58 CHF per cow per month, also economic loss from retaining unprofitable cows was about three times greater than from premature culling.
Abstract (37-40):
On average, suboptimal decisions resulted in an economic loss of 161 ± 164 CHF per farm per month (1.55 ± 1.58 CHF per cow per month), with losses from retaining unprofitable cows being approximately three times greater than those from premature culling.
Comment 2: I suggest reinforcing the scientific justification for using the Markov model, adding more references about the validity of this method in replacement decisions. It would also be useful to explain earlier in the text its relevance for sustainability.
Response:
We appreciate this comment. We have expanded the Introduction to strengthen the justification for using the Markov model, added relevant references, and explained its relevance to sustainability at an earlier point in the text.
Change in manuscript:
Introduction (66-72):
The Markov chain approach is widely recognized for its ability to model dynamic biological and economic processes in livestock systems, particularly replacement decisions, by incorporating transitions between health, reproductive, and production states over time [15–17]. Its integration here not only ensures robust economic evaluation but also allows explicit linking replacement decisions to sustainability metrics, as it quantifies the potential to extend productive lifespans and reduce environmental impacts per unit of milk.
Comment 3:
However, the economic parameters used could be described in more detail (actual values for veterinary costs, average milk prices, etc.).
Response:
We thank the reviewer for this suggestion. We have updated the Materials and Methods section to include actual average values and standard deviations for key economic parameters used in the model.
Change in manuscript:
Materials and Methods (141-148):
Economic parameters used in the model were based on farm data and reflect Swiss production conditions. Across the 29 farms, the average milk price was 0.797 ± 0.069 CHF/kg, veterinary costs averaged 977.20 ± 1080.85 CHF per cow per year, and insemination costs were 66.80 ± 20.85 CHF per service. The average cost of replacement heifers was 3123.33 ± 404.50 CHF. Feed costs were based on the prices of roughage (0.409 ± 0.12 CHF/kg DM) and concentrate feed (0.88 ± 0.26 CHF/kg DM). The average slaughter price was 3.69 ± 0.73 CHF/kg carcass weight, and calf income was calculated using an average price of 7.33 ± 2.36 CHF/kg live weight.
Comment 4: It might also be necessary to clarify any seasonal adjustments made in the model.
Response:
We thank the reviewer for raising this point. No explicit seasonal adjustments were applied in the model, as the Swiss dairy farming sector has year-round calving and does not exhibit the pronounced seasonal effects seen in some other countries.
Change in manuscript:
Materials and Methods (171-173):
Since year-round calving is common in the Swiss dairy farming sector, seasonal effects are less pronounced than in more seasonally structured systems. Therefore, no additional seasonal adjustments were applied beyond those inherently captured by the model.
Comment 5: I think some aggregated values could be complemented with measures of dispersion (SD, IQR).
Response:
We thank the reviewer for this suggestion. We have now added measures of dispersion to the main aggregated results in the Results section. Specifically, we report the replacement rate as 24.5 ± 6.9%, the average economic loss from replacement decisions as 161 ± 164 CHF per farm per month, and the corresponding value per animal as 1.55 ± 1.58 CHF per cow per month.
Change in manuscript:
In Results (196, 226-227):
Comment 6: The implications for agricultural policies or farmer support schemes could be discussed in more detail.
Response:
We agree and have expanded the Discussion to include a paragraph outlining potential policy implications.
Change in manuscript:
Discussion (362-367):
These results have direct implications for agricultural policy and farmer support schemes. Encouraging the adoption of decision-support tools, either through extension, could improve alignment between economic, ecological, and welfare outcomes. Additionally, breeding programs and herd health plans could be tailored to reduce the incidence of early culling, thereby contributing to national sustainability goals and im-proving economic resilience at the farm level.
Comment 7: The paper would be improved by discussing how these [limitations] could be overcome in future research.
Response:
We have added a sentence to the Limitations section highlighting potential approaches for addressing these issues in future studies.
Change in manuscript:
Discussion (379-382):
Such work could also address the current study’s limitations by incorporating farm-level genetic data, validating model predictions with prospective longitudinal monitoring, and expanding the dataset to include a wider range of production systems and geographical regions.
Comment 8: Add in Conclusion a recommendation to systematically integrate economic decision-support tools in farms.
Response:
We have added this recommendation to the Conclusion section.
Change in manuscript:
Conclusions (391-394):
We recommend the systematic integration of economic decision-support tools into dairy herd management, as these can help farmers identify and avoid economically suboptimal culling decisions, thereby supporting profitability, animal welfare, and environmental sustainability